# Optimizing LLM Inference Offloading with Hierarchical Scheduling and Dynamic Sparsification

## Abstract

Large language models (LLMs) power a new generation of applications. Serving them efficiently on edge remains a significant challenge due to high computational and memory costs. Current cloud-centric systems largely overlook the vast, cost-effective resources of distributed edge servers. In this paper, we introduce a novel inference offloading framework that distributes LLM workloads across a hybrid edge-cloud architecture to maximize performance and resource utilization. Our framework employs a Hierarchical Scheduling Architecture that decouples global, long-term resource planning from real-time, dynamic execution scheduling. At the kernel level, it uses Dynamic Attention Sparsification (DAS) to accelerate GPU computations by pruning redundant attention calculations. Experiments show that our hybrid approach improves overall system throughput by up to 1.86 times compared to a cloud-only baseline, effectively parallelizing workloads and introducing a scalable and robust paradigm for distributed LLM serving.

## 1 Introduction

The rapid advancement of mobile edge computing (MEC) has empowered mobile and resource-constrained devices to offload computation-intensive tasks to nearby edge servers Qu et al. (2025). By leveraging distributed processing in wireless network environments, MEC enables efficient task execution with low latency and high energy efficiency Wu et al. (2024). With the emergence of edge intelligence, the integration of artificial intelligence (AI), particularly deep learning models, into the edge computing paradigm has attracted increasing attention Dong et al. (2024). In parallel, large AI models (LAMs), including large language models (LLMs), have shown remarkable capabilities in natural language tasks. Models such as OpenAI's GPT series excel in understanding, generation, and general reasoning, owing to their massive parameter scales and powerful representational capabilities Fan et al. (2024).

Despite the tremendous potential of LLMs, their large-scale deployment encounters severe challenges. This is primarily attributed to their substantial computational and memory requirements, which hinder efficient operation in resource-constrained or latency-sensitive environments Zhou et al. (2024). Although various techniques, e.g., model compression, pruning, and distributed inference, have been proposed to address these issues, they often fall short of satisfying the real-time requirements of large-scale edge applications Wang et al. (2024). Consequently, offloading LLM tasks to appropriate edge or cloud resources has emerged as a promising approach to achieve a better trade-off among performance, latency, and resource efficiency. For instance, Yang et al. (2024) proposed PerLLM, which uses a learning-based approach to personalize scheduling and minimize energy costs. To make this offloading paradigm truly effective, especially on the resource-constrained edge, it is imperative to not only devise intelligent scheduling strategies at the system level but also to accelerate the fundamental computations at the kernel level.

Unlike traditional MEC optimization, LLMs are distinguished by their enormous parameter scales and intensive computational requirements He et al. (2024). Driven by advances in hardware capabilities and the increasing availability of large-scale datasets, researchers are continually developing models with billions of parameters. This rapid growth in model size demands substantial computational resources and high-performance infrastructure to support both training and inference pro-

cesses. The edge deployment of LLMs faces not only constraints in computational resources and memory capacity but also additional challenges related to communication overhead and multi-device coordination. As both model size and inference sequence length increase, memory consumption escalates significantly. For instance, performing single-precision (FP32) inference on LLaMA2-7B Touvron et al. (2023a;b) requires at least 28 GB of VRAM, and memory overhead increases quadratically with longer sequence lengths.

LLM inference consists of two phases: the prefill stage and the generation stage. In the prefill stage, all input tokens from the prompt are processed in parallel, similar to the forward pass in model training, resulting in high computational efficiency. In contrast, the generation stage produces tokens one by one in an auto-regressive manner, which requires significantly higher memory bandwidth and presents greater challenges for parallel execution and scheduling.

The performance objectives of task offloading for LLMs differ significantly from those of traditional computing workloads. In this study, our goal is to minimize the average inference latency of all offloaded LLM tasks while ensuring the predictive accuracy of model outputs. These objectives must be achieved under a series of practical system constraints, including available bandwidth, computational capacity, and GPU memory limitations. Against this backdrop, the rapid scaling of LLMs introduces an additional critical challenge, i.e., significantly increasing inference costs.

This paper investigates the task offloading mechanisms for LAMs/LLMs in edge computing environments, aiming to address the key challenges in scalability, efficiency, and deployment flexibility. Specifically, we explore intelligent partitioning and offloading strategies across edge–cloud hierarchies, as well as infrastructure-level optimizations to support efficient inference and fine-tuning of these models at the edge. Our contributions are as follows:

- In this work, we establish a multi-objective programming model based on model partitioning and design a hierarchical scheduling framework, which is applied to address the issues of inference offloading and resource allocation for large language models (LLMs) in edge computing environments.

- In the process of task inference, we introduce an attention sparse mask. This mask is designed to selectively filter and retain only the most critical attention weights, reducing the computational complexity caused by the dense attention mechanism while ensuring that the core semantic information required for task inference is not lost.

- We refine the FCFS scheme as a strategy for task offloading and resource scheduling in LLM inference, capturing the distinct characteristics of the prefill and decode phases by explicitly modeling the dynamics of edge computing resource allocation and task offloading latency.

- We evaluated the performance of our inference strategy using the int8 Llama2 model series. Experimental results demonstrate that our method exhibits superior and stable performance across datasets with prompts of varying lengths. Additionally, we assessed the effectiveness of network bandwidth to further validate the robustness of the proposed strategy.

## 2 SYSTEM MODEL AND PROBLEM FORMULATION

In this section, we detail the system architecture for distributed LLM inference and formulate the computational offloading problem. Our objective is to leverage limited computing resources to minimize operational costs while satisfying Quality of Service (QoS) requirements.

### 2.1 SYSTEM ARCHITECTURE

We consider a three-layer hierarchical system model. This architecture is designed to support LLM inference tasks requested by a diverse set of end-user devices.

**Cloud Server (CS):** The top tier houses a centralized cloud server with high-performance server-grade GPUs. It boasts strong computing power to handle complex, large-scale LLM inference tasks but incurs high costs.

**Edge Servers:** The middle tier comprises a set of Mobile Edge Computing (MEC) centers, denoted as $\mathcal{M} = \{M_1, M_2, \cdots, M_K\}$. Geographically distributed and user-proximal, they are equipped

with high-performance consumer-grade GPUs. Though individually less powerful than data center counterparts, they are abundant and more cost-effective for specific workloads.

**End-User Devices:** The bottom tier consists of heterogeneous end-user devices, denoted as $\mathcal{U} = \{U_1, U_2, \cdots, U_N\}$, including PCs and smartphones across diverse scenarios. They generate LLM inference requests.

We model an LLM inference request $T_i$ from user $U_i$ as a task characterized by its model type and the number of tokens to generate. The key decision is to determine where to offload this task. Let $x_{ij}$ denote the offloading decision, where $j \in \mathcal{M} \cup \mathrm{CS}$. We define $x_{ij} = 1$ if task $T_i$ is offloaded to server $j$, and $x_{ij} = 0$ otherwise.

## 2.2 PROBLEM FORMULATION

The challenge of efficient LLM inference offloading is formulated as a multi-goal programming problem. The primary objective is to devise an optimal task allocation strategy that maximizes throughput to the greatest extent possible while minimizing the total network transmission latency across all tasks. This is achieved by intelligently distributing inference requests between a central cloud server and a set of geographically distributed edge servers, subject to the inherent physical constraints of the edge infrastructure and the stringent Quality of Service (QoS) requirements of the inference tasks.

The primary objective is to devise an optimal task allocation strategy that:

- **Maximizes system throughput**, i.e., the number of inference tasks successfully processed within their QoS constraints, and

- **Minimizes total network transmission latency**, i.e., the sum of round-trip latencies for all offloaded tasks.

The multi-objective optimization problem is defined as:

$$\min \quad \{S(t, x_{ij}, \delta_{ij}), \ T(x_{ij}, L_{ij}^{\mathrm{trans}})\}, \tag{1}$$

$$\text{s.t.} : \sum_{j \in \mathcal{S}} x_{ij} = 1, \quad \forall i \in \{1, \cdots, N\}, \tag{1a}$$

$$\sum_{k=1}^{K} y_{jk} M_k \leq S_j, \quad \forall j \in \mathcal{M}, \tag{1b}$$

$$x_{ij} \leq y_{jk}, \quad \forall i \in \{1, \cdots, N\}, \forall j \in \mathcal{M}, \tag{1c}$$

$$\sum_{i=1}^{N} x_{ij} W_i \leq \Omega_j, \quad \forall j \in \mathcal{S}, \tag{1d}$$

$$L_{ij}^{trans} + L_{ij}^{comp} \leq L_{max}, \quad \forall i, j, \tag{1e}$$

where $\mathcal{T} = \{T_1, \cdots, T_N\}$ is the set of $N$ inference tasks, indexed by $i$. $\mathcal{S} = \mathcal{M} \cup \{\mathrm{CS}\}$ is the set of all available servers, including the subset of edge servers $\mathcal{M}$ and the central server CS, indexed by $j$. $t$ indicates the generation time of tokens. $x_{ij}$ is the primary binary decision variable, where $x_{ij} = 1$ if task $T_i$ is offloaded to server $j$, and $x_{ij} = 0$ otherwise. $S(t, x_{ij}, \delta_{ij}) = -\frac{1}{t} \sum_{i=1}^{N} \sum_{j \in \mathcal{S}} x_{ij} \cdot \delta_{ij}$, $T(x_{ij}, L_{ij}^{\mathrm{trans}}) = \sum_{i=1}^{N} \sum_{j \in \mathcal{S}} x_{ij} \cdot L_{ij}^{\mathrm{trans}}$. Indicator $\delta_{ij}$ variable that equals 1 if task $T_i$ can be completed by server $j$ within its QoS deadline, 0 otherwise. $L_{ij}^{trans}$ is the parameter representing the round-trip transmission latency for the input and output data of task $T_i$ when processed by server $j$.

For Constraints 1b and 1c, $\mathcal{K} = \{C_1, \cdots, C_K\}$ is the set of $K$ unique computing tasks, indexed by $k$. $M_k$ is the storage size of the uncompressed $C_k$. $S_j$ is the available storage capacity of edge server $j$. $y_{jk}$ is a secondary binary decision variable, where $y_{jk} = 1$ if $C_k$ is deployed on edge server $j$, and $y_{jk} = 0$ otherwise. $mod(i)$ is a function mapping task $T_i$ to the specific computing tasks $k \in \mathcal{K}$ it requires. For Constraint 1d, $W_i$ is the computational workload of task $T_i$, $\Omega_j$ is the total available computational capacity of server $j$ over a given period. For Constraint 1e, where $x_{ij} = 1$, $L_{ij}^{comp}$ is

the computation latency for server $j$ to execute task $T_i$. $L_{max}$ is the maximum tolerable end-to-end latency, defining the required Quality of Service.

This is achieved by intelligently distributing inference requests between a central cloud server and a set of geographically distributed edge servers, while respecting the physical resource limitations of edge infrastructure and the stringent QoS constraints of inference workloads.

# 3 INFERENCE OFFLOADING OPTIMIZATION STRATEGY

In this section, we first describe the hierarchical scheduling architecture, which represents the macro-level strategy for system-wide resource management. Then, we elaborate on Dynamic Sparse Attention Acceleration, our micro-level strategy to reduce the computational burden on individual nodes, particularly edge servers. Finally, we present the Optimization Offloading Strategy that integrates both levels for offloading and resource allocation of LLMs inference tasks in cloud-edge networks, as shown in Fig. 1.

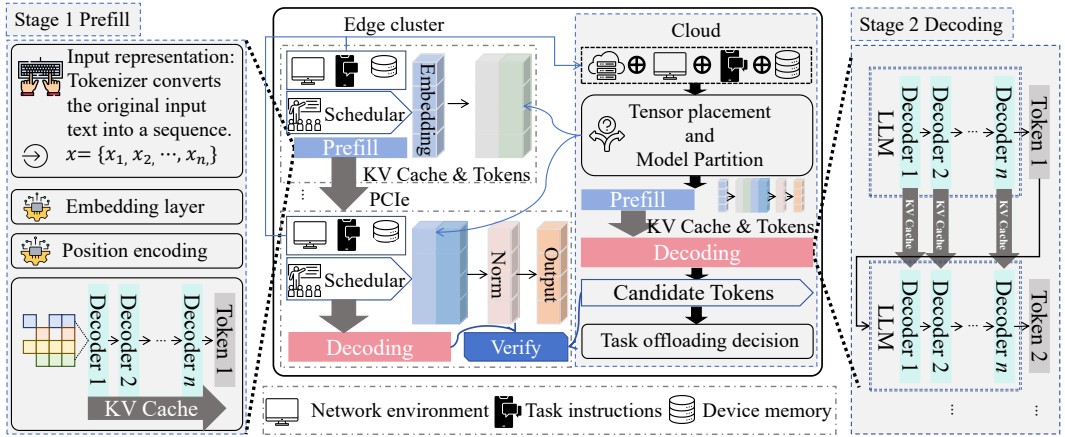

Figure 1: Architecture of the inference framework. The framework divides the reasoning process into two stages, the prefill stage and the decoding stage, and leverages the collaboration between edge devices and the cloud to improve performance.

## 3.1 HIERARCHICAL SCHEDULING ARCHITECTURE

The complexity of LLM inference necessitates a sophisticated resource management strategy. This section proposes a hierarchical scheduling architecture designed to decouple long-term strategic resource decisions from real-time tactical execution planning.

### 3.1.1 CENTRALIZED ADAPTIVE TENSOR PLACEMENT

This phase is responsible for global, long-term strategic resource allocation decisions. It is deployed on a central server GPU cluster, which possesses substantial computational power and a comprehensive view of the entire distributed hardware ecosystem. Its primary task is to characterize system-wide hardware resources and analyze model parameters to generate an optimal static model deployment blueprint for subsequent online inference.

The central server GPU cluster first conducts a detailed measurement of all available hardware resources, including Memory capacity $C_{mem}^{(d)}$ for each device $d \in \mathcal{D}$ (GPU/CPU). Computational throughput $T_{comp}^{(d)}$ for each device $d$. Interconnect bandwidth $B_{inter}^{(d_1,d_2)}$ between any two devices $d_1, d_2$ (e.g., PCIe or RDMA).

These hardware specifications, along with the configurations of the target model $M_T$ and draft model $M_D$ (including their parameter sets $\mathcal{P}_T, \mathcal{P}_D$ and computational graphs), are fed into the Adaptive Tensor Placement module. This module solves a global optimization problem to determine the initial static allocation of model parameters (tensors) across the heterogeneous memory hierarchy.

Let $x_{p,d} \in \{0, 1\}$ be a binary decision variable indicating whether tensor $p$ is placed on device $d$. To minimize the anticipated data transfer overhead during the online phase, we define $Acc(p)$ as an indicator function, which denotes whether the tensor $p, p \in \mathcal{P}_T \cup \mathcal{P}_D$ is accessed by both devices $d_1, d_1 \in \mathcal{D}$ and $d_2, d_2 \in \mathcal{D}, d_1 < d_2$ simultaneously.

$$\min_X \sum_p \sum_{d_1} \sum_{d_2} Acc(p) \cdot x_{p,d_1} \cdot (1 - x_{p,d_2}) \frac{\text{Size}(p)}{B_{inter}^{(d_1,d_2)}}. \tag{2}$$

$$\text{s.t.} : \sum_{p \in \mathcal{P}_T \cup \mathcal{P}_D} x_{p,d} \cdot \text{Size}(p) \leq C_{mem}^{(d)}, \quad \forall d \in \mathcal{D}, \tag{2a}$$

$$\sum_{d \in \mathcal{D}} x_{p,d} = 1, \quad \forall p \in \mathcal{P}_T \cup \mathcal{P}_D. \tag{2b}$$

According to Constraint 2a, the allocated memory on each device must not exceed its total capacity. In Constraint 2b, each tensor must be placed on exactly one device. The output of this phase is a global tensor placement map $X^* = x_{p,d}$, which is disseminated to individual edge server GPUs.

### 3.1.2 DECENTRALIZED DYNAMIC PIPELINE PLANNING

This phase is responsible for scheduling inference execution. It is primarily deployed on individual edge server GPUs, which directly receive and process batched inference requests. Each edge server dynamically generates a fine-grained execution plan for the current batch, based on its local hardware configuration and the tensor placement strategy provided by the central server.

Upon receiving a batched inference request, the allocator on an edge server GPU utilizes the following information. The relevant portion of the global tensor placement map $X^*$ is pre-computed during the offline phase. The local hardware capabilities of the current edge device, such as its own memory capacity $C_{mem}^{(edge)}$ and computational throughput $T_{comp}^{(edge)}$. Characteristics of the incoming batch, e.g., sequence length $L$ and batch size $N$. The allocator dynamically computes an interleaved pipeline execution plan, specifying critical scheduling parameters.

To minimize the end-to-end inference latency for the current batch. Let the decision variables be the target model prefill micro-batch size, $\mu_{pref} \in \mathbb{Z}^+$. Target model decode micro-batch size, $\mu_{dec} \in \mathbb{Z}^+$. Draft model batch size: $b_{draft} \in \mathbb{Z}^+$. Number of candidate tokens to generate in each speculative decoding step $k_{cand} \in \mathbb{Z}^+$.

$$\min_{\mu_{pref}, \mu_{dec}, b_{draft}, k_{cand}} \mathcal{L}(N, L, \mu_{pref}, \mu_{dec}, b_{draft}, k_{cand}), \tag{3}$$

$$\text{s.t.} : \mu_{pref} \leq N, \quad \mu_{dec} \leq N, \quad b_{draft} \leq N, \tag{3a}$$

$$\text{Req}(\mu_{pref}, \mu_{dec}, b_{draft}, k_{cand}, L) \leq C_{mem}^{(edge)}. \tag{3b}$$

where $\mathcal{L}(\cdot)$ is a comprehensive function that estimates the total delay based on batch characteristics $(N, L)$, scheduling parameters, and system resources (e.g., $X^*, C_{mem}^{(edge)}, T_{comp}^{(edge)}$), while implicitly incorporates computational and memory bandwidth bottlenecks. Constraint 3b indicates that the memory requirements for the KV cache, intermediate activations, and local tensor segments of the edge GPU must not exceed its local memory capacity. DAS addresses the challenge of executing tasks on resource-constrained edge servers faster and more efficiently.

## 3.2 DYNAMIC ATTENTION SPARSIFICATION

DAS is a method that accelerates sparse Flash Attention Zhang et al. (2025) by dynamically filtering redundant computations within the attention mechanism itself. Attention operations are often the primary computational bottleneck, and attention maps in many models exhibit structured sparsity, where semantically similar tokens show high resemblance. We introduce binary block masks $M_g$ and $M_{pv}$.

**Definition 1** *For query, key, and value blocks $Q_i, K_j, V_j$, and attention weights $P_{e_{ij}} = softmax(Q_i K_j^\top / \sqrt{d_k})$. If $M_g[i, j] = 0$, the block matrix multiplication $Q_i K_j^\top$ is skipped. If $M_{pv}[i, j] = 0$, the computation of $P_{e_{ij}} V_j$ is additionally skipped, given that $Q_i K_j^\top$ has been computed.*

DAS employs a two-stage online filtering mechanism to generate these masks and accelerate the process:

### 3.2.1 ADAPTIVE BLOCK PRUNING

This stage aims to generate task $M_g$ by predicting the sparse structure, thereby avoiding the computation of the full attention map.

For blocks $B_k$ in $Q$ and $K$ (i.e., $Q_i$ or $K_j$), if their internal self-similarity is high, which is measured by cosine similarity above a threshold $\theta_B$, the entire block is compressed into a single representative token by averaging. A compressed attention map $\hat{P}$ is then efficiently computed from these representative tokens.

$$\hat{P}_{ij} = \text{softmax}(\hat{Q}_i \hat{K}_j^\top / \sqrt{d_k}). \tag{4}$$

A cumulative distribution function Pianosi & Wagener (2015) is applied to select the most significant attention score blocks in $\hat{P}$, whose cumulative sum reaches a threshold $\tau$. Crucially, to preserve vital information, computations involving blocks with low internal self-similarity (i.e., less than $\theta_B$) are never skipped.

$$M_g[i,j] = \begin{cases} 1 & (i,j) \in S_{TC} \cup S_P, \\ 0 & \text{otherwise.} \end{cases} \tag{5}$$

where $S_{TC}$ denotes the set of block index pairs $(i,j)$ selected by the cumulative distribution function, while $S_P$ denotes the set of block index pairs $(i,j)$ where the internal similarity of the query or key does not exceed the threshold $\theta_B$. This mask $M_g$ dictates skipping the full $Q_i K_j^\top$ computation.

### 3.2.2 ONLINE VALUE PRUNING

Following the initial block filtering via $M_g$, this stage employs a sparse online Softmax method to prune computations further. The specific generation logic for $M_{pv}$ is determined by the sparse online Softmax mechanism, whose objective is to identify and prune value matrix multiplications with negligible impact on the final output. It generates the mask $M_{pv}$, used to skip the $P_{e_{ij}} V_j$ multiplication for blocks that, even if computed, would have a negligible contribution to the final output.

$$M_{pv}[i,j] = \begin{cases} 1 & V_j = f(P_{e_{ij}}), \\ 0 & \text{otherwise.} \end{cases} \tag{6}$$

where $f(.)$ means $P_{e_{ij}}$ has significant contribution to $V_j$.

## 3.3 OPTIMIZATION OFFLOADING STRATEGY

### 3.3.1 THROUGHPUT-OPTIMAL OFFLOADING PRINCIPLE

An offloading strategy is deemed to achieve a specific throughput rate $\lambda$ if it can maintain stability of the system-wide request queue under this task arrival rate. In the parlance of queuing theory, this implies that the underlying discrete-time Markov chain (DTMC) Li et al. (2025) representing the system state is irreducible and positive recurrent.

**Definition 2 (Work-Conserving Offloading)** *An offloading strategy $\pi$ is defined as work-conserving if it never leaves a capable compute resource (either the Central Server (CS) or an Edge Server (ES), i.e., an MEC server $M_k \in \mathcal{M}$) idle, provided there exists at least one pending inference task in the global request queue that that specific resource can process.*

It prioritizes the immediate assignment of a pending task to an available server rather than, for instance, delaying batch requests for a specific server type while other resources remain unutilized.

### 3.3.2 THE K-PRIORITY FCFS OFFLOADING ALGORITHM

While the work-conserving principle offers a high-level guideline, its practical implementation necessitates a concrete algorithmic framework. Anticipating the use of stability analysis, we introduce the K-Priority First-Come-First-Serve (K-P-FCFS) family of offloading algorithms.

Let $K_q \geq 1$ be an integer representing the look-ahead window. The central scheduler operates on the global queue of incoming requests, denoted as $Q$.

**Limited Decision Space:** At each scheduling instance, the scheduler restricts its decision space by considering only the $K_q$ oldest requests residing in $Q$. This limitation is pragmatic, as it bounds the computational overhead of the scheduler itself, preventing it from becoming an internal bottleneck within the Sparge-Offload architecture.

**Prioritization Rule:** Among these $K_q$ selected requests, a specific prioritization rule is applied. This rule can range from a simple First-Come-First-Served (FCFS) Zhao & Stankovic (1989) discipline to more sophisticated heuristics.

**Task Assignment:** The highest-priority request is then assigned to an available server $j \in \mathcal{M} \cup \{\text{CS}\}$ that satisfies all task-specific constraints, required model availability, and maximum permissible latency for the specific request type.

This framework is inherently work-conserving (as long as a valid assignment is made whenever possible) and sufficiently flexible to incorporate various optimization heuristics and real-world operational constraints.

## 4 EXPERIMENT

This section presents a comprehensive evaluation of our proposed method within a collaborative edge computing environment. We first outline the experimental setup, including the testbed, benchmark models, datasets, and workloads, as well as baseline methods for comparison and evaluation metrics. Subsequently, we present and analyze the overall performance of the method against the baselines in terms of inference latency and throughput.

### 4.1 EXPERIMENTAL SETUP

#### 4.1.1 ENVIRONMENT AND RESOURCE LIMITATION

Our experiments were conducted on a physical testbed composed of five heterogeneous computational devices. The hardware configuration includes four edge servers, each equipped with an NVIDIA GeForce GTX TITAN X GPU, and a central cloud server. The cloud server is equipped with an NVIDIA RTX 4090 GPU. All devices are interconnected via a router and a switch, with a physical bandwidth of 50 Mbps. To emulate variable network conditions found in real-world deployments, we employed the Linux TC utility Hubert et al. (2002) to dynamically adjust the network bandwidth and communication latency between devices. Due to the limited VRAM, we use the int8 precision model for inference in all the following experiments.

#### 4.1.2 BENCHMARKS AND DATASETS

To validate the effectiveness and scalability of our approach, we selected a suite of state-of-the-art open-source LLMs from the Llama2 family Touvron et al. (2023a;b), specifically, Llama2-7B, and Llama2-13B. To thoroughly assess performance across diverse tasks and prompt length distributions, we utilized several well-established LLM benchmark datasets. These datasets cover a range of tasks, including code generation, knowledge-based question answering, and text summarization:

**HumanEval Chen et al. (2021)**: An OpenAI dataset comprising 164 handwritten programming problems to evaluate the model's code generation capabilities.

**C-Eval Huang et al. (2023)**: A comprehensive Chinese evaluation suite containing 13,948 multiple-choice questions across various subjects, designed to test the model's Chinese language understanding and reasoning abilities.

**SummEval Fabbri et al. (2021)**: A dataset focused on news text summarization, which includes 100 articles from the CNN/DailyMail corpus and their corresponding professional summaries.

**WikiText-2 Merity et al. (2016)**: A language model benchmark dataset that contains over 100 million primitives extracted from high-quality Wikipedia articles. Specifically, we extract the subset of samples where the length of input tokens is 32 and the number of generated tokens is 96.

### 4.1.3 EVALUATION METRICS

We employed two key metrics to measure the performance of each method:

**Inference Latency:** Measured as the total time elapsed from the user's request to the reception of the complete generated output, in milliseconds (ms/token). Lower latency indicates better performance.

**Throughput:** Defined as the number of tokens the system can process per second (tokens/s). Higher throughput signifies greater processing capacity. To measure throughput, the batch size was set to the maximum supported by the participating devices.

## 4.2 PERFORMANCE COMPARISON

To establish a comprehensive performance benchmark, we evaluated all methods on the text generation task using the Llama2-7B and Llama2-13B models. The end-to-end inference latency and throughput results are summarized in Table 1.

| Model | Configuration | HumanEval | | C-Eval | | SummEval | | WikiText-2 | |
|-------|--------------|---------|-----------|---------|-----------|---------|-----------|---------|-----------|
| | | Latency | Throughput | Latency | Throughput | Latency | Throughput | Latency | Throughput |
| | Edge Only | 97.90 | 10.09 | 92.18 | 5.54 | 104.79 | 4.41 | 88.24 | 7.74 |
| Llama2-7B | Cloud Only | 43.96 | 25.09 | 41.08 | 20.94 | 47.90 | 18.50 | 39.22 | 23.31 |
| | Edge + Cloud | 57.94 | 36.45 | 54.61 | 32.09 | 60.88 | 29.07 | 51.96 | 34.82 |
| | Edge Only | 259.74 | 6.45 | 245.49 | 1.90 | 279.44 | 0.72 | 235.29 | 4.21 |
| Llama2-13B | Cloud Only | 104.90 | 10.14 | 99.20 | 5.68 | 111.78 | 4.32 | 95.10 | 7.94 |
| | Edge + Cloud | 134.87 | 18.84 | 128.26 | 14.53 | 141.72 | 12.63 | 122.55 | 17.07 |

Table 1: Performance of LLM inference, Latency: ms/token, Throughput: tokens/s.

Cloud-only outperforms edge-only setups. For Llama2-13B on HumanEval, cloud-only reduces latency by 60% (259.74 ms to 104.90 ms) and increases throughput by 57% (6.45 to 10.14 tokens/s). High-performance GPUs exhibit low-latency performance in the absence of communication overhead. The hybrid Edge + Cloud architecture maximizes throughput via parallel processing, outperforming both cloud-only and edge-only across models. For Llama2-7B on HumanEval, it boosts throughput by 45% (25.09 to 36.45 tokens/s), for Llama2-13B, the gain reaches 86% (10.14 to 18.84 tokens/s), demonstrating effective workload distribution. Hybrid systems incur latency trade-offs due to communication overhead. For Llama2-13B, hybrid latency is 28% higher than cloud-only (134.87 ms vs. 104.90 ms), attributable to network communication synchronization between edge and cloud. This underscores a key trade-off: parallelization enhances throughput but increases single-task response time via communication costs.

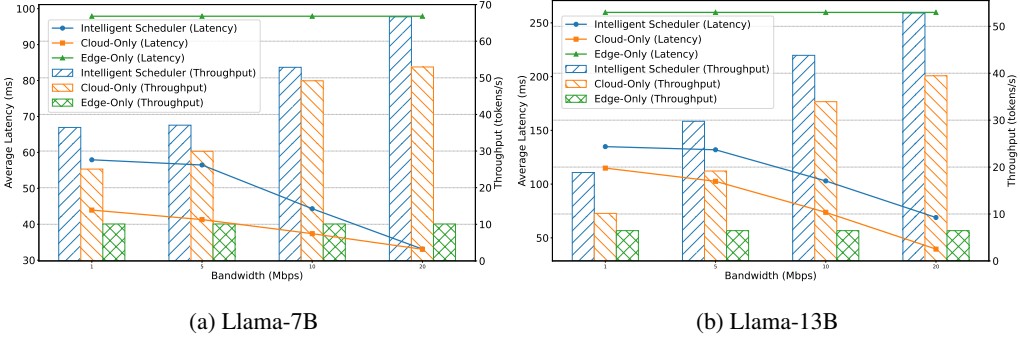

(a) Llama-7B  (b) Llama-13B

Figure 2: Impact of Bandwidth to Collaborative LLMs Inference of Llama-7B and Llama-13B

Fig. 2 evaluates the performance of edge-cloud collaborative inference for Llama-7B and Llama-13B models under varying bandwidth conditions, focusing on average latency and throughput. For both models, the Intelligent Scheduler (our work) consistently outperforms the Cloud-Only and Edge-Only baselines.

As illustrated in Fig. 3, the cloud-only baseline devotes resources to both prefilling and decoding, causing prolonged server occupancy with each draft-verify cycle. In contrast, Edge + Cloud dedicates the server to adaptive resource allocation while offloading inference tasks to more cost-effective edge devices, reducing server runtime by approximately 40% in all model configurations.

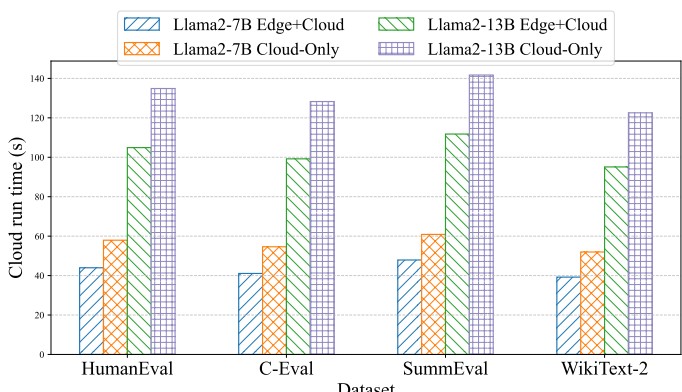

Figure 3: Average cloud run time for a task between Cloud-only and Edge + Cloud.

### 4.3 ABLATION STUDY

To verify the impact of each module on performance, an ablation study was conducted to compare the latency and throughput across different component combinations.

| DA | FA | Sche | Auto | Dec | Ran | Latency | Throughput |
|----|----|------|------|-----|-----|---------|------------|
|    | ✓  |      | ✓    |     | ✓   | 145.52  | 8.13       |
| ✓  |    |      | ✓    |     | ✓   | 78.15   | 10.22      |
| ✓  |    | ✓    |      |     | ✓   | 72.3    | 11.89      |
| ✓  |    |      | ✓    | ✓   |     | 65.71   | 18.55      |
| ✓  |    | ✓    |      | ✓   |     | 57.94   | 36.54      |

Table 2: Ablation study of proposed techniques on HumanEval dataset based on Llama2-7B. Dynamic Attention Sparsification (DA), with impact on latency; Full-Attention (FA), a standard full-attention baseline without sparsification; Hierarchical Scheduling Architecture (Sche); Autoregressive Model (Auto), a baseline for autoregressive decoding that generates one token at a time; K-Priority FCFS Offloading Algorithm (Dec), an intelligent priority-based offloading algorithm; and Random Offloading (Ran), a naive offloading baseline with the worst expected performance. Latency: ms/token, Throughput: tokens/s.

As shown in Table 2, the results indicate that Dynamic Attention Sparsification (DA) significantly outperforms Full-Attention (FA): under the Auto+Ran configuration, latency is reduced by 46% (from 145.52 ms to 78.15 ms) and throughput is increased by 26% (from 8.13 to 10.22 tokens/sec), confirming its effectiveness in enhancing computational efficiency. Hierarchical Scheduling Architecture (Sche) requires collaboration with intelligent strategies: when combined with K-Priority FCFS Offloading (Dec), compared to the Auto+Dec configuration, latency decreases by 12% (from 65.71 ms to 57.94 ms) and throughput nearly doubles (from 18.55 to 36.54 tokens/sec), reflecting the pipeline gains of system-level scheduling. Dec significantly outperforms Random Offloading (Ran): under the DA+Sche configuration, throughput rises by 206% (from 11.89 to 36.54), demonstrating the necessity of intelligent offloading.

## 5 CONCLUSION

In this paper, we propose a comprehensive and efficient framework for LLM inference offloading, designed to maximize throughput in hybrid edge-cloud environments. Our approach involves implementing a Hierarchical Scheduling Architecture to manage system-wide resources from a macro perspective, decoupling strategic, offline tensor placement from real-time, online execution planning. Simultaneously, we accelerate individual GPU computations from a micro perspective through Dynamic Attention Sparsification (DAS), a technique that prunes redundant attention calculations. The entire system is guided by a throughput-optimal, work-conserving offloading principle, which we theoretically prove to guarantee system stability under heavy load. Experimental results on Llama2-7B and Llama2-13B models demonstrate that our hybrid approach significantly boosts system throughput, validating the effectiveness of our proposed framework.

## ETHICS STATEMENT

This research was conducted in full alignment with the ICLR Code of Ethics. We affirm that our study did not involve human participants or animal subjects. The datasets utilized, including HumanEval, C-Eval, etc, were procured from public sources and handled in strict accordance with their respective licensing and usage terms, ensuring no breach of privacy. Our methodology was designed to be impartial and to prevent discriminatory outcomes, and the research did not involve personally identifiable information (PII). We are fully committed to upholding the principles of research integrity and transparency throughout this work.

## REPRODUCIBILITY STATEMENT

To facilitate the reproducibility of our research, we provide comprehensive resources for independent verification. The source code implementing our primary contributions is included as a Supplementary Material.zip file in the supplementary material submitted with this manuscript. This package contains a self-contained implementation of our core innovations Furthermore, we offer a thorough explanation of our core contribution, to support independent implementation.

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

## LLMs USAGE DISCLOSURE

The authors disclose that Large Language Models (LLMs) were used only to aid or polish writing.

## A APPENDIX

### RELATED WORK

In the context of Large Language Model (LLM) inference, the prefill and decode stages Pan et al. (2024) exhibit distinct computational characteristics: the former is compute-bound, primarily involving the batch processing of input sequences, while the latter is bandwidth-bound, focusing on token-by-token generation and KV Cache access Raiaan et al. (2024). Addressing these differing computational demands, two main paradigms currently exist: the Fusion Paradigm and the Separation Paradigm.

In contrast to the Fusion Paradigm, the Separation Paradigm leverages the computational differences between prefill and decode by attempting to distribute them across distinct computational devices. Systems such as Splitwise Choukse et al. (2025), DistServe Zhong et al. (2024), and Mooncake Qin et al. (2025) are representative examples of this approach, having inherited and further developed the concept of separation. The primary advantage of the Separation Paradigm lies in its ability to configure independent levels of parallelism for both the prefill and decode stages, offering greater system flexibility. However, the principal challenge encountered by the Separation Paradigm is the efficient transfer of KV Cache between disparate devices. This necessitates high-speed network interconnects within the cluster, and the associated network costs are substantial.

### CONSTRAINT

To achieve the objective function 1, we formulate a set of constraints.

**Constraint 1 (Task Assignment Uniqueness)** *This constraint ensures that every task is executed exactly once by a single server in the network.*

$$\sum_{j \in \mathcal{S}} x_{ij} = 1, \quad \forall i \in \{1, \cdots, N\}. \tag{7}$$

*This is a fundamental logical constraint that guarantees the integrity of the solution. It ensures that every task $T_i$ in the set is processed while preventing redundant executions of the same task, which would waste both computational and network resources.*

**Constraint 2 (Edge Server Model Placement & Storage)** *This critical set of constraints addresses the core challenge of deploying LLMs on storage-limited edge servers.*

$$\sum_{k=1}^{K} y_{jk} M_k \leq S_j, \quad \forall j \in \mathcal{M}, \tag{8}$$

$$x_{ij} \leq y_{j,mod(i)}, \quad \forall i \in \{1, \cdots, N\}, \forall j \in \mathcal{M}, \tag{9}$$

*where $\mathcal{K} = \{C_1, \cdots, C_K\}$ is the set of $K$ unique computing tasks, indexed by $k$. $M_k$ is the storage size (e.g., in GB) of the uncompressed $C_k$. $S_j$ is the available storage (e.g., RAM/VRAM) capacity of edge server $j$. $y_{jk}$ is a secondary binary decision variable, where $y_{jk} = 1$ if $C_k$ is deployed on edge server $j$, and $y_{jk} = 0$ otherwise. $mod(i)$ is a function mapping task $T_i$ to the specific computing tasks $k \in \mathcal{K}$ it requires.*

Eq. equation 8 models the physical storage limitation of each edge server. It dictates that the sum of the sizes of all models deployed on an edge server cannot exceed its capacity, $S_j$. Eq. equation 9 creates a crucial logical link between storage limitation and task assignment. It states that a task $T_i$ can be assigned to an edge server $j$ ($x_{ij} = 1$) only if the specific computational procedure required for that task, $model(i)$, is already deployed on server $j$ ($y_{j,model(i)} = 1$).

**Constraint 3 (Server Computational Capacity)** *The total computational demand placed on any server must not exceed its processing capabilities.*

$$\sum_{i=1}^{N} x_{ij} W_i \leq \Omega_j, \quad \forall j \in \mathcal{S}, \tag{10}$$

*where $W_i$ is the computational workload of task $T_i$, for instance, measured in Floating Point Operations (FLOPs). $\Omega_j$ is the total available computational capacity of server $j$ over a given period.*

This constraint addresses the "limited compute" aspect of edge servers. Even if a model can be stored, the server must have the power to execute the inference task efficiently. This inequality prevents the assignment of an excessive number of tasks to any single server, particularly less powerful edge nodes, which would result in queuing delays and high computational latency. It effectively enforces load balancing based on the heterogeneous compute capacities across the system.

**Constraint 4 (End-to-End Latency Quality of Service)** *The total latency for any executed task, combining both network transmission and on-device computation, must not exceed a predefined service-level threshold.*

$$L_{ij}^{trans} + L_{ij}^{comp} \leq L_{max}, \quad \forall i, j, \tag{11}$$

*where $x_{ij} = 1$, $L_{ij}^{comp}$ is the computation latency for server $j$ to execute task $T_i$. $L_{max}$ is the maximum tolerable end-to-end latency, defining the required Quality of Service.*

This constraint is paramount for ensuring a satisfactory user experience. While the objective function (1) focuses on minimizing transmission latency, this constraint provides a hard upper bound on the total perceived latency. It prevents the model from choosing a server that, despite being geographically close (low $L_{ij}^{trans}$), is too slow or heavily loaded to perform the computation promptly (high $L_{ij}^{comp}$). This ensures that every task assignment in the final solution is not only network-efficient but also meets the strict performance criteria required for real-time applications.

ACCURACY PRESERVATION

A critical consideration in any optimization involving model alterations is the preservation of task accuracy. Our framework ensures high accuracy through two primary mechanisms.

First, regarding the use of int8 precision models, as mentioned in Section 4.1.1, we utilize the official quantized versions of the Llama2 models. These models are generated using robust post-training quantization (PTQ) techniques that are widely validated to have a negligible impact on accuracy for the benchmark tasks evaluated in this paper, such as HumanEval and C-Eval.

Second, our proposed Dynamic Attention Sparsification (DAS) is explicitly designed to be accuracy-aware. Unlike naive pruning, DAS employs a principled, two-stage filtering process to identify and remove only computationally redundant attention heads. The thresholds $\theta B$ (for internal self-similarity) and $\pi$ (for cumulative score) are critical hyper-parameters that can be tuned to balance performance and accuracy. For our experiments, these thresholds were carefully selected to ensure that no significant degradation in accuracy was observed on downstream tasks. Specifically, our pruning strategy preserves vital information by never skipping blocks with high informational entropy (i.e., low self-similarity) and retaining the most significant attention scores, thereby maintaining the core semantic processing capabilities of the model.

The core principle behind the accuracy preservation of Dynamic Attention Sparsification (DAS) is that the pruned computations contribute negligibly to the final output of the attention layer. We can formalize this by bounding the error introduced by the sparsification process.

Let $O_{\text{full}} \in \mathbb{R}^{N \times d_v}$ be the output of a standard, full attention layer for a sequence of $N$ tokens, and let $O_{\text{DAS}} \in \mathbb{R}^{N \times d_v}$ be the output from our DAS-enabled attention layer. The error introduced by DAS can be represented as the norm of their difference, $E = \|O_{\text{full}} - O_{\text{DAS}}\|$. Our goal is to show that $E$ is bounded by a small value controlled by our pruning hyperparameters.

The output of the attention layer is computed block-wise. Let $Q_i, K_j, V_j$ be the $i$-th query block and $j$-th key/value blocks, respectively. The full output is:

$$O_{\text{full}} = \sum_{i,j} \text{softmax}\left(\frac{Q_i K_j^T}{\sqrt{d_k}}\right) V_j = \sum_{i,j} P_{ij} V_j \tag{12}$$

where $P_{ij}$ is the attention probability matrix between block $i$ and $j$.

Our DAS framework introduces two binary masks, $M_g$ (Adaptive Block Pruning) and $M_{pv}$ (Online Value Pruning). The output of DAS is:

$$O_{\text{DAS}} = \sum_{i,j} M_g[i,j] \cdot M_{pv}[i,j] \cdot P_{ij} V_j \tag{13}$$

The approximation error is the sum of the contributions from the pruned blocks:

$$E = \left\| \sum_{(i,j) \in \mathcal{S}_{\text{pruned}}} P_{ij} V_j \right\| \leq \sum_{(i,j) \in \mathcal{S}_{\text{pruned}}} \| P_{ij} V_j \| \tag{14}$$

where $\mathcal{S}_{\text{pruned}} = \{(i,j) \mid M_g[i,j] \cdot M_{pv}[i,j] = 0\}$. We can analyze the error contribution from each pruning stage.

**1. Bounding Error from Adaptive Block Pruning** ($M_g$). This stage prunes blocks based on a cumulative distribution function applied to a compressed attention map $\hat{P}$, with a threshold $\tau$. This means we keep the blocks that are predicted to have the highest attention scores. The sum of the scores of the pruned blocks is therefore bounded. Let $\mathcal{S}_g$ be the set of blocks pruned by this stage. We have:

$$\sum_{(i,j) \in \mathcal{S}_g} \|\hat{P}_{ij}\|_1 \leq 1 - \tau \tag{15}$$

Assuming the compressed attention map $\hat{P}$ is a reasonable proxy for the full attention $P$, i.e., $\|P_{ij}\|_F \leq C\|\hat{P}_{ij}\|_1$ for some constant $C$ related to block size, the error contribution $E_g$ from this stage can be bounded. Let $V_{\max} = \max_j \|V_j\|_F$.

$$E_g = \sum_{(i,j) \in \mathcal{S}_g} \|P_{ij} V_j\|_F \leq \sum_{(i,j) \in \mathcal{S}_g} \|P_{ij}\|_F \|V_j\|_F \leq V_{\max} \sum_{(i,j) \in \mathcal{S}_g} \|P_{ij}\|_F \leq C \cdot V_{\max}(1-\tau) \tag{16}$$

This shows that the error introduced by block pruning is directly controlled by $(1-\tau)$. By choosing $\tau$ close to 1 (e.g., $\tau = 0.99$), this error component is kept small.

**2. Bounding Error from Online Value Pruning** ($M_{pv}$). This stage prunes the $P_{ij} V_j$ computation for blocks that passed the first stage but whose attention scores in $P_{ij}$ are determined to be negligible. The sparse online softmax algorithm ensures that for any pruned element $(m,n)$ within a block $(i,j)$, its value $(P_{ij})_{mn}$ is below a small threshold $\epsilon_{pv}$. Let $\mathcal{S}_{pv}$ be the set of blocks pruned at this stage. For any $(i,j) \in \mathcal{S}_{pv}$, all elements of $P_{ij}$ are small. Let the block size be $b \times b$.

$$\|P_{ij}\|_F = \sqrt{\sum_{m,n=1}^{b} (P_{ij})_{mn}^2} < \sqrt{b^2 \cdot \epsilon_{pv}^2} = b \cdot \epsilon_{pv} \tag{17}$$

The error contribution $E_{pv}$ from this stage is then bounded:

$$E_{pv} = \sum_{(i,j) \in \mathcal{S}_{pv}} \|P_{ij} V_j\|_F \leq V_{\max} \sum_{(i,j) \in \mathcal{S}_{pv}} \|P_{ij}\|_F < |\mathcal{S}_{pv}| \cdot V_{\max} \cdot b \cdot \epsilon_{pv} \tag{18}$$

This error is proportional to the small threshold $\epsilon_{pv}$.

**Conclusion.** The total error is bounded by the sum of errors from both stages:

$$\|O_{\text{full}} - O_{\text{DAS}}\| \leq E_g + E_{pv} \leq \mathcal{O}(1-\tau) + \mathcal{O}(\epsilon_{pv}) \tag{19}$$

This demonstrates that the error is directly controlled by the hyperparameters $\tau$ and $\epsilon_{pv}$. By selecting a high value for $\tau$ and a low value for $\epsilon_{pv}$, we can ensure that the output of the DAS-enabled attention layer remains arbitrarily close to that of the full attention layer, thus preserving the model's accuracy.

PROOF SKETCH FOR SYSTEM STABILITY

In Section 3.3, we claim our work-conserving offloading principle can achieve throughput-optimality. This claim is rooted in queuing theory, specifically leveraging concepts from Lyapunov stability analysis. Here, we provide a high-level proof sketch.

1. **System Model:** We model the system as a discrete-time queueing network. The state of the system at time slot $t$ is defined by the vector of queue lengths $Q(t) = [Q_{CS}(t), Q_{M_1}(t), \ldots, Q_{M_K}(t)]$, where $Q_j(t)$ is the number of pending tasks for server $j \in \mathcal{S} = \{CS, M_1, \ldots, M_K\}$. Let $\lambda_i$ be the arrival rate of tasks of type $i$. The total arrival rate vector is $\boldsymbol{\lambda}$.

2. **Capacity Region:** The system has a capacity region $\Lambda$, which is the set of all arrival rate vectors $\boldsymbol{\lambda}$ for which there exists some scheduling policy that can keep all queues stable (i.e., the expected queue lengths remain bounded over time).

3. **Lyapunov Function:** To prove stability, we define a quadratic Lyapunov function, which represents the total squared backlog in the system:

$$L(Q(t)) = \frac{1}{2} \sum_{j \in \mathcal{S}} Q_j(t)^2 \tag{20}$$

4. **Lyapunov Drift:** We analyze the one-step expected change (drift) in the Lyapunov function, conditioned on the current state $Q(t)$:

$$\Delta(Q(t)) = \mathbb{E}[L(Q(t+1)) - L(Q(t)) \mid Q(t)] \tag{21}$$

A key result from queuing theory (based on the Foster-Lyapunov theorem) states that if we can find a policy that ensures the drift is negative whenever the queue length is large, for any arrival rate $\boldsymbol{\lambda}$ strictly inside the capacity region $\Lambda$, then the system is stable.

5. **Work-Conserving Policy:** Our K-P-FCFS offloading algorithm is a **work-conserving** policy. This means it never leaves a server idle if there is a compatible task that it can process. For any arrival rate vector $\boldsymbol{\lambda} \in \Lambda$, there exists some $\epsilon > 0$ such that the expected service rate vector $\boldsymbol{\mu}$ can satisfy $\mu_j > \lambda_j + \epsilon$ for all servers $j$. A work-conserving policy, by maximizing service opportunities, aims to satisfy this condition.

Under such a policy, the Lyapunov drift can be shown to satisfy the following inequality:

$$\Delta(Q(t)) \leq B - \epsilon \sum_{j \in \mathcal{S}} Q_j(t) \tag{22}$$

where $B$ is a positive constant that depends on the second moments of the arrival and service processes, and $\epsilon > 0$. For a sufficiently large total queue length ($\sum Q_j(t) > B/\epsilon$), the negative term dominates, forcing the drift to be negative. This guarantees that the queue lengths are bounded and the system is stable.

This result establishes that our policy is **throughput-optimal** in the sense that it can stabilize the system for any arrival rate that is theoretically serviceable, i.e., any $\boldsymbol{\lambda}$ strictly within the capacity region $\Lambda$.

