# OpenReview forum: "Optimizing LLM Inference Offloading with Hierarchical Scheduling and Dynamic Sparsification"
_ICLR.cc/2026/Conference — ICLR 2026 Conference Withdrawn Submission_

### Official Review · Reviewer_en61 · 2025-10-26

**Soundness:** 1
**Presentation:** 1
**Contribution:** 1
**Rating:** 0
**Confidence:** 4

**Summary:**

This paper studies the task of offloading LLM inference to edge servers. It proposes three avenues for improving throughput and latency. One is Dynamic Sparse Attention which skips certain computations in the MHA block. The second is a centralized tensor placement module that makes decisions about where to place the model and request memory blocks, provided full visibility into all memory, computational, and networking resources. Last is a request assignment algorithm framework. This system is tested on a cluster with 5 GPUs, 50 Mbps interconnects, and 2 models from the Llama2 family.

**Strengths:**

I'd like to thank the authors for submitting their work. This is an interesting area of research with ongoing demand due to the proliferation of  Generative AI.

**Weaknesses:**

Unfortunately, I don't believe this work is in a state to be published. There are 4 fundamental flaws that need to be addressed in the paper before it can be reconsidered, and they are:
* The authors introduce Dynamic Attention Sparsification (DAS), which is a ***lossy compression technique***. DAS skips certain computations in the original LLM, and ***will change the LLM output text, potentially drastically***. When such changes are proposed, it is common to evaluate on downstream applications, e.g., test DAS-equipped LLM on Q&A, GSM8K, SWEBench, or even a simple perplexity score. As this is absent from the paper, DAS's effects on the LLM's accuracy can not be considered negligible.
* In the evaluation section in §4.2, the headline experiment is comparing the latency and throughput of the proposed system vs. ***only cloud*** and ***only edge***. This is not fair, as the proposed system has more hardware to use. At minimum, ***the system should be compared with the aggregated inference system comprised of separate cloud-based + separate edge-based***. In fact, judging based on the numbers in Figure 2, this fair baseline would do as well as the proposed system.
* There is ***no reasonable discussion of related work*** in the paper, except for short snippets of the text in §1. The research area of LLM inference has increasingly many research papers, many of which may be similar and need discussing. Without a related work section, it is difficult to say this paper is novel. The paper doesn't even discuss the most famous edge-inference or dual-cluster collaboration papers like FlexGen [1] or SplitWise [2]. FlexGen could actually be a reasonable baseline as it already considers different memory hierarchies within a single node, which can be extended to multiple nodes in a straightforward process.
* The biggest flaw is the presentation of the work. I would highly recommend the authors ask third-parties to read the paper and ask for feedback. Here are some detailed suggestions for rewriting:
  * In §1, the authors do not justify why edge computing is beneficial for LLM applications? LLMs need powerful GPUs, their computational runtime far outweighs network transmission times (each token in a llama3-70B model, served on H100s, using two-way tensor parallelism, takes 20ms, and each request involves 100s of tokens, compared to network RTTs that are at absolute worst 200ms).
  * §2 begins with a detailed mathematical representation of a problem that has not been explained yet. The dense descriptions in §2 can all be moved to the appendix, and the saved space can be spent on explaining the overall architecture by providing intuition, examples, design justifications, etc.
  * §3.3 is supposed to describe how the request scheduling works, but ends up talking about a framework without providing actual details: "Among these Kq selected requests, a specific prioritization rule is applied. This rule can range from a simple First-Come-First-Served (FCFS) Zhao & Stankovic (1989) discipline to more sophisticated heuristics."
  * Figure 1 has too much unnecessary detail and is hard to follow. Consider removing less critical visuals and focus on providing an overview of the system without going into the detail of every module.
  * In evaluation in §4.1, the compared inference software is not mentioned. What is the inference engine (commonly vLLM, SGlang, TensorRT-LLM, ...) or orchestration stack (NVIDIA Dynamo, llm-d, AIBrix, ...)?

Some other minor comments that do not affect the decision:
* Please use `citep` for generating citations. Almost all citation are formatted incorrectly.
* There are numerous punctuation and grammatical errors. The text would benefit from proofreading.

[1] Sheng, Ying, et al. "Flexgen: High-throughput generative inference of large language models with a single gpu." International Conference on Machine Learning. PMLR, 2023.

[2] Patel, Pratyush, et al. "Splitwise: Efficient generative llm inference using phase splitting." 2024 ACM/IEEE 51st Annual International Symposium on Computer Architecture (ISCA). IEEE, 2024.

**Questions:**

Please see the weaknesses section.

---

### Official Review · Reviewer_RHYr · 2025-10-28

**Soundness:** 2
**Presentation:** 3
**Contribution:** 2
**Rating:** 2
**Confidence:** 3

**Summary:**

The paper proposes a hybrid edge–cloud framework for large language model (LLM) inference that integrates (1) a Hierarchical Scheduling Architecture for multi-tier resource management and (2) a Dynamic Attention Sparsification (DAS) technique to reduce GPU-level computation. The system formulates LLM inference offloading as a multi-objective optimization problem, aiming to minimize latency and maximize throughput under resource and QoS constraints. Experiments on Llama-2 7B and 13B models show throughput improvements of up to 1.86× over cloud-only baselines.

**Strengths:**

The paper targets an important and timely scenario—efficient large language model (LLM) inference in hybrid edge–cloud environments. This direction is practically relevant as LLM deployment increasingly extends beyond centralized data centers. The proposed hierarchical scheduling idea provides a structured view of separating global resource allocation from local execution, and the introduction of Dynamic Attention Sparsification (DAS) aligns with recent interest in combining system-level and kernel-level optimizations. The work also attempts to formalize the joint optimization problem under latency and resource constraints, and the experimental framework, though limited, demonstrates an integrated prototype that covers scheduling, computation, and evaluation.

**Weaknesses:**

- **The experimental setup does not represent realistic edge environments.** The so-called edge devices are equipped with TITAN X GPUs—hardware much closer to workstation-class GPUs than to mobile or embedded processors. The models used (Llama2-7B and 13B in int8 precision) fit comfortably into these GPUs, so the experiments do not expose actual resource bottlenecks or demonstrate the claimed feasibility of running LLMs on low-end edge hardware. This mismatch between the claimed scenario and the evaluated testbed severely limits the empirical significance.

- **The evaluation is also limited by the absence of competitive baselines.** The paper compares only simple Edge-only, Cloud-only, and Edge + Cloud configurations. The authors should include more recent systems that tackle collaborative edge–cloud scheduling with proven designs. Without such comparisons, the reported throughput improvements provide little insight into real advantages.

- **Network bandwidth plays a central role in determining system latency and throughput, yet it is not explicitly modeled in the optimization or scheduling framework.** While bandwidth is varied empirically in the experiments, it should appear as a key decision variable or constraint to capture its first-order effect on task placement and communication overhead. The lack of bandwidth-aware modeling limits the generality and realism of the proposed approach.

- **The framework further assumes a static set of inference tasks, ignoring the stochastic nature of real-world request arrivals.** In practice, LLM serving workloads follow bursty or heavy-tailed distributions, and queueing dynamics heavily influence end-to-end latency. By not incorporating temporal or stochastic effects, the analysis oversimplifies the real deployment setting.

**Questions:**

- The paper positions itself as an edge–cloud collaborative inference framework, but the discussion of related work is rather limited. Could the authors provide a more complete comparison with recent edge–cloud collaborative LLM serving systems? What are the current state-of-the-art results in this area, and how does the proposed method differ conceptually or empirically from these systems?

- The multi-objective optimization formulation is given in symbolic form, but there is no evidence of how it is solved or applied. Is there an implemented solver or heuristic policy for this optimization? Are the results shown in the experiments derived from the optimization output, or are they pre-configured settings? Showing quantitative optimization outcomes (e.g., Pareto trade-off curves) would greatly clarify this part.

- The claimed improvement of “up to 1.86× throughput compared to a cloud-only baseline” appears to be an unfair comparison, because the hybrid setting uses one cloud GPU plus four additional edge GPUs. Could the authors normalize the throughput by the total GPU count, compute cost, or energy usage, to show per-resource efficiency? Under such normalization, what is the actual throughput gain?

- Since bandwidth plays a crucial role in edge–cloud collaboration, why is it not explicitly modeled in the optimization or scheduling framework? Would bandwidth-aware modeling change the offloading decisions?

- Real-world LLM serving workloads typically exhibit stochastic or bursty request arrivals rather than fixed task sets. How would the proposed scheduling and throughput-optimality analysis extend to such dynamic workloads? Have the authors tested queue stability or latency distribution under random arrivals?

---

### Official Review · Reviewer_ofg8 · 2025-10-29

**Soundness:** 2
**Presentation:** 1
**Contribution:** 2
**Rating:** 4
**Confidence:** 3

**Summary:**

This paper tackles the challenge of serving large language model (LLM) inference workloads across hybrid edge–cloud infrastructures. The authors propose (i) a hierarchical scheduling architecture that separates global tensor placement from per-batch execution planning, (ii) a Dynamic Attention Sparsification (DAS) technique that prunes attention computations via block-level masks, and (iii) a K-priority FCFS offloading heuristic that aims to maintain queue stability. Empirical evaluation on a small-scale testbed (four TITAN X edge GPUs plus one RTX 4090 cloud GPU) with int8 LLaMA2 variants suggests throughput improvements over edge-only and cloud-only baselines, with additional ablations examining the proposed modules.

**Strengths:**

Clarity: The paper is readable overall, and the decomposition of the proposed system into macro (scheduling) and micro (kernel-level sparsification) components is conceptually helpful.
Significance (potential): Improving LLM inference in hybrid environments is important, and combining scheduling with kernel optimizations is a relevant goal.

**Weaknesses:**

* Originality: The hierarchical scheduling plus adaptive tensor placement approach closely resembles prior art (e.g., DistServe Zhong et al. 2024, Splitwise Choukse et al. 2025, Mooncake Qin et al. 2025). The motivation and formalization overlap heavily, yet direct comparisons or clear differentiators are missing. Similarly, DAS appears to be a straightforward extension of sparse FlashAttention (Zhang et al. 2025) with heuristic masking; no new theoretical insight or novel sparsity pattern is articulated.
* Baselines are weak. The comparison excludes state-of-the-art hybrid or speculative decoding schedulers (e.g., DistServe, SARATHI). There is no evaluation against other attention sparsification methods (Longformer variants, recent block-sparse kernels), leaving the incremental benefit unclear.
* The system-scale study is limited to four decade-old TITAN X GPUs connected via 50 Mbps links. This setting is far removed from realistic MEC deployments or modern edge accelerators; consequently, claims about scalability or cost-effectiveness are insufficiently supported.
* Key measurements such as end-to-end accuracy, queue backlog, or energy consumption are absent. Since DAS prunes attention, verifying fidelity is essential; likewise, queue stability claims should be corroborated via stress tests (e.g., varying arrival rates).
Clarity (missing details):
* The tensor placement stage uses binary decisions with a single placement per tensor (Constraint 2b), which conflicts with the need to replicate weights across devices for load balancing. This requires clarification.
* The ablation table labels (DA/FA/Sche/Auto/Dec/Ran) are terse; the exact combinations evaluated are difficult to parse, and only HumanEval results are shown.

**Questions:**

* Novelty relative to existing systems: How does the hierarchical scheduler differ concretely from Splitwise, DistServe, or Mooncake, which already partition prefill/decode stages and perform joint scheduling? Can you provide algorithmic distinctions and direct empirical comparisons?

* Queue stability validation: Beyond the proof sketch, have you simulated or measured queue lengths under varying arrival rates to demonstrate stability? If so, please report those metrics; if not, could you extend the experiments to do so?

* Scalability and hardware realism: Why were legacy TITAN X GPUs chosen as edge devices? How would the approach perform on modern edge accelerators (e.g., A10, L4) or with higher-bandwidth (5G/ethernet) links? Please discuss the bottlenecks and provide at least simulated evidence.

* Comparison with alternative sparsification: How does DAS compare against existing block-sparse or top-k attention mechanisms (e.g., FlashAttention-3 with dynamic sparsity) in terms of speedup and accuracy? A head-to-head would strengthen the contribution.

* Tensor placement constraint: Given Constraint (2b), how do you handle the need for multiple replicas of widely reused tensors across devices? Is partial replication or sharding considered? If not, does this constraint limit achievable throughput?

---

### Official Review · Reviewer_872x · 2025-10-30

**Soundness:** 3
**Presentation:** 3
**Contribution:** 3
**Rating:** 6
**Confidence:** 3

**Summary:**

This paper presents a framework for optimizing LLM inference in hybrid edge–cloud environments through a hierarchical scheduling architecture and Dynamic Attention Sparsification (DAS). The system jointly optimizes tensor placement, resource scheduling, and computation sparsity to improve throughput and latency. The formulation and idea is quite interesting.

**Strengths:**

- Good formulation and theoretical setup: The multi-objective optimization model and the throughput optimality proof via Lyapunov analysis are rigorous, while not particularly novel.
- Architecture design: Decoupling global tensor placement from local scheduling is interesting and practical for distributed inference.
- Throughput and latency results: Results on Llama-2 (7B, 13B) show up to 1.8× throughput improvement with clear ablations that isolate each component’s benefit.

**Weaknesses:**

- Novelty is a bit limited since the ideas are based on paradigms like DistServe or Splitwise but tthe innovation is their integration and formulation.
- Scale of experiments: Larger models and higher number of GPUs in experiments can strengthen the work.
- Comparisons with other scheduling methods are missing in the experimental section.

**Questions:**

Can you have comparisons with other baselines?

---

### Note · Authors · 2025-11-23

I have read and agree with the venue's withdrawal policy on behalf of myself and my co-authors.